# Differences in Body Composition and Maturity Status in Young Male Volleyball Players of Different Levels

**DOI:** 10.3390/jfmk8040162

**Published:** 2023-11-23

**Authors:** Alessia Grigoletto, Mario Mauro, Stefania Toselli

**Affiliations:** 1Department of Biomedical and Neuromotor Science, University of Bologna, Via Selmi 3, 40126 Bologna, Italy; alessia.grigoletto2@unibo.it; 2Department of Life Quality Studies, University of Bologna, 47921 Rimini, Italy

**Keywords:** sports, anthropometric characteristics, body composition, maturity status, young volleyball players

## Abstract

Volleyball is an intermittent team sport that requires specific anthropometrical and physical characteristics for winning performance. The present study aimed to evaluate the maturity status of the young male players of eight volleyball teams, and to observe differences in anthropometric characteristics and body composition. Ninety-four male adolescent volleyball players were recruited during a national tournament carried out in Treviso (Italy). Anthropometric characteristics such as weight, stature, skinfold thicknesses, circumferences and diameters, and bioelectrical impedance were measured. The biological maturation was estimated for all players. Each team was classified as a higher or lower lever according to its tournament ranking. A two-way ANOVA compared team levels and players’ maturity status. Considering the maturity offset, 62 boys were classified as “on time”, 20 as “late”, and 12 as “early”. Three clubs presented many boys with “early” as the maturity offset, and two of these finished the tournament in the first position. Young volleyball players classified as “early” seemed to show anthropometric characteristics linked to better performance at the tournament (higher height, upper arm and calf muscle area, fat mass percentage, and total fat-free mass). The results of the present study could have practical implications for talent selection, but further studies are needed to better evaluate the effect of maturity status on the characteristics of volleyball players.

## 1. Introduction

Volleyball is an intermittent sport that requires high-intensity performance of an intermittent nature, i.e., frequent short bouts of high-intensity exercise followed by periods of low-intensity activity and brief rest periods [1,2,3]. Suitable anthropometric and body composition characteristics and high technical and tactical skills are needed to succeed in this sport [4,5]. The frequent jumps that are usually performed during a volleyball match require specific characteristics, such as thinness and explosive muscle power. Among anthropometric variables, leg length, arm span, and height differ between high-level players, along with physical skill, such as coordination in agility tests and vertical jumps [6,7]. Height, arm span, and upper and lower body power have been identified as key factors for performance in both male and female adolescent volleyball players [8,9]. However, few studies have discussed volleyball players’ physical and functional characteristics, particularly during adolescence. In addition, the available literature principally focuses on female volleyball players [10,11], but there are far fewer studies on males.

Regarding adolescence, the influence of maturity status on physical and physiological characteristics has attracted increased scientific interest, considering its relevance for sports performance. Biological maturation can be defined as the timing and tempo of progress to achieving a mature state [12]. The physical development of young players is strongly influenced by maturity status, especially as regards their body composition and physical capacities [13,14,15].

Understanding the role of maturity in physical characteristics and performance in youth athletes during adolescence is essential, since this period coincides with the selection of players. Sport is selective, chiefly during adolescence, and often occurs along a maturity-related gradient. Many studies have analyzed the influence of maturity status on physical, physiological, and performance characteristics in soccer, basketball, or handball players [13,14,16,17,18,19], but less information exists on male volleyballers. Albaladejo-Saura and colleagues reported that volleyball players with a more advanced state of maturation exhibited higher values of height, arm span, sitting height, bone diameters, muscle perimeters and fat, muscle and bone masses, and better performance achieved in medicine ball throwing and in countermovement jump (CMJ) than their chronological age peers [20]. Since variables such as height, sitting height, leg length, and muscle circumference have a high correlation with performance in physical fitness tests related to volleyball requirements, the best values obtained by volleyball players with an advanced maturity status testify how this state represents a competitive advantage in the sport performance of volleyball during adolescence.

To our knowledge, no previous studies were carried out about bioelectrical impedance vectorial analysis (BIVA) and young volleyball players. Therefore, the present study aims to (a) compare the prevalence of maturity status among volleyball players of the teams that have reached different positions in the ranking of a national tournament, and (b) investigate the relationship between maturity status and anthropometric, performance, and body composition parameters and BIVA. These two aspects are strongly connected with talent selection.

It was hypothesized that players who reached a higher position in the ranking would exhibit differences in maturity status and their anthropometric and body composition profile. In particular, people with an early maturation could have better results in the final racking, and they could show higher values for some anthropometric characteristics, such as stature, circumferences, and lower value of fat mass in comparison with boys classified as on time or with a late maturation.

## 2. Materials and Methods

### 2.1. Participants and Study Design

This is an observational study assessed between the 17th and 18th of June 2022, during the National Tournament “0.13 Torneo Città di Treviso”, organized in Treviso (Italy) from the volleyball society Volley Treviso. Eight teams of 22 were randomly selected to be measured during the study: Volley Treviso, La Piave Volley, Kosmos Volley, Pallavolo Sestese, Cisanonembro’thers, Gas Sales Bluenergy Piacenza, Virtus Fano, and VT Personal Time. A total of 94 young male volleyball players were evaluated (Volley Treviso: 11, La Piave Volley: 12, Kosmos Volley: 11, Pallavolo Sestese: 12, Cisanonembro’thers: 9, Gas Sales Bluenergy Piacenza: 12, Virtus Fano: 13, VT Personal Time: 14). Figure 1 shows the study design. All the evaluations were assessed within a Treviso sports center where a private room was set up for specific environmental features such as a temperature between 22 °C and 24 °C and air humidity between 50 and 60%.

The volume of the weekly workouts of each team was collected from all coaches, and each player trained for about 6 h per week (four workouts of 90 min each). In each training unit, 45 min was spent on strength and conditioning and coordinative capabilities, whereas 45 min was spent on technical–tactical skills. No diet information was collected.

Participants were informed and volunteered to decide to participate in the study. Their parents were informed and provided written consent. This study was in accordance with the Declaration of Helsinki and approved by the Bioethics Committee of the University of Bologna (N. prot. 25027).

### 2.2. Anthropometry

A trained operator collected all the anthropometric measurements, such as weight, height, circumferences, and skinfold thickness, according to standardized procedures [21]. The mean value of three measurements was gathered. Weight was measured to the nearest 0.1 kg using a calibrated analogue scale. Height and sitting height were collected at the nearest 0.1 cm using a stadiometer (GPM, Zurich, Switzerland). The body mass index (BMI) was calculated as the ratio between weight (kg) and squared stature converted in meters (m).

Circumferences (relaxed and contracted upper arm, waist, hip, calf) were measured to the nearest 0.1 cm with a non-stretchable tape. The upper arm circumference was taken on the subject in a standing position, at the mid-point between the shoulder acromion and the olecranon process point, with the participant’s elbow relaxed along the body side (stretched evaluation) or to be flexed 90◦ with palm facing upward (contracted evaluation); the waist circumference was taken on the subject in a standing position with close feet and arm along the trunk, at the minimum abdominal circumference line, between the inferior margin of the last rib and the iliac crest. The hip circumference was taken on the subject in a standing position with close feet and arms along the trunk, at the highest point of the glutes; the calf circumference was taken at the bulkiest calf point, with the participant in a standing position (calf muscles stretched).

Diameters (humerus and femur) were taken to the nearest 0.1 cm with a sliding caliper, both on the left side of the body. The humerus and femoral widths were taken, respectively, between the own lateral and medial epicondyles, with the participant’s elbow and knee flexed 90°.

Skinfold thicknesses (biceps, triceps, subscapular, supraspinal, suprailiac, thigh, medial, and lateral calf) were measured to the nearest 1 mm using a Lange skinfold caliper on the left side of the body (Beta Technology Inc., Houston, TX, USA) at the following sites: triceps and biceps, vertically at the midpoint between the acromion process and the olecranon process, respectively, at the posterior and anterior upper arm face; subscapular, at an angle of 45″ to the lateral side of the body, about 20 mm below the tip of the scapula; suprailiac, about 20 mm above the iliac crest (in the axillary line); supraspinal, about 20 mm above the iliac spine; calf, vertically at the bulkiest calf point both medially and laterally.

Then, body composition parameters such as fat-free mass (FFM), fat mass (FM), and percentage of fat mass (%F) were estimated according to the equation developed by Slaughter et al. [22]. According to Frisancho’s equations, many body areas were estimated, such as the total area of the upper arm (TUA) and of the lower limb (TCA), muscle area of the upper arm (UMA) and lower limb (CMA), and fat area of the arm (UFA) and lower limb (CFA) [23]. In addition, calf and arm fat indexes (FCI and UFI) were derived.

### 2.3. Maturity Status

Mirwald and colleagues developed a specific equation for boys to estimate the years from the peak height velocity (PHV), which is an important index of adolescent growth [24]. Maturity offset represents the time before or after the PHV; by subtracting the age at PHV from chronological age, it is possible to estimate the year from PHV.
MO=−9.236+0.0002708(leg length∗sitting height)−0.001663(age∗leg l007216(age∗sitting height)+0.02292(weight/height).

Children who are not yet in their adolescent growth spurt often have a lower approximation of the age at PHV (APHV) and those who have already passed their adolescent growth spurt are often higher [12]. For this reason, age-specific Z-score was used to classify the young athletes. Based on the age-specific standardized Z-score of the predicted APHV, boys were classified as later (Z > 1), on time (−1.0 ≤ Z ≤ 1.0), and earlier Z < 1.0 maturing [25].

### 2.4. Bioelectric Impedance Vector Analysis (BIVA)

Bioelectric impedance analysis (BIA) was used to measure impedance. An electric current was used with a frequency of 50 kHz (BIA 101 BIVA^®^ PRO, Akern, Florence, Italy). The participants were in the supine position, with four electrical conductors; two electrodes were placed on the right hand and two on the right foot after cleaning the skin with alcohol [26,27]. Subjects were asked to put their lower limbs at an angle of 45° compared to the median line of the body and to put their upper limbs at an angle of 30° from the trunk. Athletes received the instruction to abstain from foods and liquids for ≥4 h before the test. BIVA was carried out using the classic methods, e.g., normalizing R (Ω) and Xc (Ω) for height in meters [28]. Both the elite male volleyball players’ and the general adolescent male population’s bioelectrical-specific ellipses were used as a reference to build the 50%, 75%, and 95% tolerance ellipses on the R/H–Xc/H graph. BIVA plots the parameters recorded in BIA (R, Xc, PhA) as a vector within a specific tolerance ellipse (specific profile for each sport and competitive level), and it allows the evaluation of soft tissues through patterns based on percentiles of their electrical characteristics. A BIVA vector that falls out of the 75% tolerance ellipses exhibits a different tissue impedance compared to the selected reference population, while vectors that fall in the 50% ellipse represent common impedance characteristics.

### 2.5. Statistical Analysis

The eight teams were divided into two groups (higher level, HL; lower level, LL) according to their final ranking at the tournament (teams that reached at least quarterfinals = HL, teams that lost before quarterfinals = LL). The mean and standard deviation (SD) of the two groups were calculated for each variable and the frequency of appearance (percentage) was determined for the maturity status. The distribution of the variables’ residuals was verified with the Shapiro–Wilk test. When a variable presented a right-skewed curve, the logarithm transformation was applied to meet the normality distribution assumption. The two-tailed one-way analysis of variance (ANOVA) was performed to evaluate the differences between the two groups and among maturity statuses. When a variable’s distribution could not meet the normality assumption, a non-parametric statistic test was performed (Mann–Whitney rank-sum and Kruskal–Wallis’s rank tests). The probability of the type-I error was settled at <0.05. Finally, a post hoc Tukey evaluation was used to evaluate the difference between the final position at the tournament and between the maturity status when the Snedecor–Fisher statistical test probability value (*F*) was observed as significant.

## 3. Results

Table 1 shows the maturity status prevalence according to the tournament’s final ranking. Three teams were classified as higher-level due to the results of the tournament, and five teams were classified as lower level. Teams with a worse ranking presented a higher number of boys with later maturity status, whereas the ratio of players who matured on time was similar (HL = 69.44%, LL = 63.79%).

Table 2, Table 3 and Table 4 show the mean and standard deviation of each variable for both the ranking group and the maturity status, and it reports the statistical comparisons between them and their interaction.

Regarding the differences linked to the ranking position, better teams exhibited significantly higher values in leg length and femoral diameter and lower amounts of fat on the most informative skinfolds and fat percentage. On the contrary, boys who stopped before the quarterfinals showed significantly higher values in arm circumference, arm and calf skinfold thicknesses, and fat area or percentage on their lower and upper limbs (TUA, UFA, UFI, CFA, and CFI). Also, players clustered in the HL group showed a wider skeletal robustness in their lower limbs (femoral diameter).

Several statistically significant anthropometric differences were relative to maturity status. Boys classified as early showed better values in many important anthropometric characteristics such as height, weight, all the circumferences, and calf muscle area, and body composition parameters such as fat mass and fat-free mass than on-time and later youths.

Finally, regarding the interaction effect between ranking and maturity status, the earlier young players classified as higher-level showed significantly wider values in height, leg length, and femoral diameter than the earlier young players classified as lower level. In addition, the earlier boys ranked between the lower level presented higher values in parameters related to the local (triceps, subscapular, supraspinal, suprailiac, and lateral calf skinfolds, UFI, CFI) and total body fat mass (%F, FM) than earlier players classified in the first positions. Finally, although players who matured on time showed better characteristics in HL than LL teams in body composition (%F, FM), the LL players were taller and exhibited longer low limbs.

### Bioimpedance Vector Analysis (BIVA)

Figure 2 and Figure 3 show BIVA results regarding both the final ranking of the tournament (on the left) and the maturity status (on the right).

Figure 2 shows significant differences in BIVA vector distance according to the final ranking (Figure 2A) and between the boys classified as early and late (Figure 2B).

Figure 3 shows different vector placements in the ellipses in accordance with the reference population. Compared to the general adolescent reference population (Figure 3A), only the boys who matured in an average manner were included in the 50% tolerance, while the early-matured boys exhibited a lower level of biological electric resistance. The early boys belonging to winning teams showed a wider displacement compared to leaner cell mass (left size vector position). In addition, they had a body composition more akin to the elite population of male adult volleyball players (Figure 3B). Differently, players of the HL teams who matured in an average manner or later exhibited the greatest BIVA differences compared to elite volleyball players (Figure 3B, blue triangle and diamond), especially in hydration and lean mass. As regards LL teams, all the maturity categories showed wide displacement against both the general adolescent population and the elite volleyball reference group. However, they were closer to the adolescent reference population than the adult elite volleyball players.

## 4. Discussion

The present study had two aims: (a) to compare the prevalence of maturity status among volleyball players of the teams that reached different positions in the ranking of a national tournament and (b) to investigate the relationship between maturity status and anthropometric and body composition parameters and BIVA. Our beginning hypotheses speculate that players who reached a higher position in the final ranking would exhibit differences in maturity status and their anthropometric profiles. Also, we believe that players who mature earlier show better body composition.

Many studies have been performed regarding the influence of maturity status on the body, physical performance, and physiological characteristics on the growing and scouting of adolescent soccer, basketball, or handball players, while less information exists on male volleyballers. The elite players have rapidly increased their physical demands in recent years, and, for this reason, recruiters and coaches put greater emphasis on physical fitness, and talent selection, from an early age [16,26]. In fact, in recent years, the identification of adolescent talent has gained increased interest from both the scientific community and sports managers [29]. The implementation of early talent identification programs could bring advantages to the teams that carry them out, both in economic and sporting terms [30].

Regarding the prevalence of maturity status, in the present study, significant differences were observed in the boys classified as late-maturing in comparison with those who were early or on time. In the teams that achieved higher ranking positions, only four boys were classified as late, while in the teams ranked between the lower levels, there were sixteen of them. This is in accordance with previous studies that demonstrated that maturity status has an important role in performance in adolescent males [21,31,32,33,34]. In fact, Romeo-Garcia and colleagues found that young male handball athletes who presented an early biological maturation achieved higher values in anthropometric characteristics and in physical tests [18]. They observed significant differences in basic measurements, such as weight, height, fat-free mass, BMI, and Cormic Index, and in some physical tests, such as medicine ball throw and squat jump, with the group of early maturers, who had the highest values. On the contrary, Toselli and colleagues did not find any differences in maturation category prevalences between elite and non-elite adolescent soccer teams from 11 to 14 years old [16]. However, having boys classified as late in the team reduces the possibility of winning and of demonstrating good performance in a short time. Despite this, the immediate advantage of premature maturation may not be associated with great future performance and talent expression. The role of coaches and trainers is fundamental for enhancing and scouting hidden talents.

According to the above-mentioned results, we found that teams that did not reach the quarterfinals showed higher values in several parameters linked to body fat and worse body composition. In fact, they exhibited higher values in several skinfold thicknesses, in body fat percentage, and in the fat area of the limbs. These results are in accordance with a previous study that investigated the effect of team level, maturation, and interaction in adolescent soccer players [16]. Many fat-related parameters differed between elite and non-elite players such as triceps, biceps, subscapular, suprailiac, and thigh skinfold thicknesses, and arm, thigh, and calf fat indexes. However, both young and adult volleyball players must make explosive movements and they may be powerful, agile, and rapid; for this reason, low body fat is required, particularly for young volleyball players [35]. Teams classified between the higher levels showed significantly higher values in leg length and femoral diameter, which are two important characteristics in volleyball. Height and leg length are fundamental in volleyball due to the height of the net (2.43 m for elite volleyball players, 2.15 m in U-13 competitions) [36].

Regarding differences due to maturity status, the present results are in line with a study conducted by Albaladejo-Saura et al. [20]. The authors found higher values in several anthropometric characteristics (such as height, diameters, trunk height, etc.), in volleyball players with a more advanced state of maturation, akin to what emerged in the present study. Among the anthropometric characteristics, the greatest differences between the two groups were found for skinfolds. This is in line with previous studies regarding soccer, which showed the importance of monitoring body fat, since appropriate levels of fat permit the players to move more effectively during training and games [37,38].

Regarding the results of the BIVA graphs, it is interesting to notice that early boys classified in the first positions had a body composition like the elite population of volleyball players, showing a lower level of resistance and leaner body. In addition, their vector characteristics differed against the general adolescent reference population. The premature growth of the muscle cells and the reduction of the inactive mass (fat mass) are relevant parameters in fast and power sports such as volleyball [2,28]. This could explain the better performance of these teams and could also be an important factor to consider and monitor the BIVA parameter changes over time for talent selection. At the same time, it is interesting to note that the boys in teams classified at lower levels had a similar position in both the BIA vector graphs, independent of maturity status. The boys classified on time and in the first position were plotted out from the tolerance ellipse of the elite volleyball players’ population. This could be justified by the maturity status because they are near the PHV, which is a moment of big changes for the body. Also, this information could confirm that maturation in adolescence could widely affect changes in anthropometry and body composition, impacting physical performance and team scouting. Although only seven players out of thirty-six were classified as early-maturing in high-level teams, volleyball involves six players on the court for any action and two boys having improved body and physical characteristics could lead to winning.

Previous studies reported that the chance of selection for relatively younger soccer players was widely affected by maturation status, physical performance, and anthropometric characteristics, whereas relatively older athletes had a selection advantage independent of their maturity status [39,40]. It is difficult to provide an exhaustive comparison, but it seems that the influence of this aspect is the same in this sport.

The present study has several limitations. The study design included only one period of evaluation and longitudinal research with several follow-ups could enrich the specific literature. The teams were randomized and selected to be measured during the tournament, and it was not possible to evaluate all the teams involved. It could have been interesting to measure all the teams participating in the tournament to have a wider sample size and to collect more data for maturation state comparison. Also, the participants were only thirteen-year-old males; many investigations considering both sexes and different ages are suggested. In addition, it was not possible to collect information about the diet habits of the young male volleyball players. No data were given about the years of experience of the players, which could influence the final ranking, or about the time on the court of each player. Finally, physical tests (for example, jumping test or speed test) were not performed and no data related to match results and skills were collected. Future investigations could draw more complete study designs in order to evaluate the correlations between physical performance, match analysis, anthropometry and body composition, match level, and biological maturation.

## 5. Conclusions

In conclusion, in the present study, young male volleyball players classified as early had higher values of the anthropometric characteristics linked to better performance (represented by the final ranking of the tournament). In fact, among the eight teams, two of them that presented the most early maturing boys were ranked in the top places of the tournament (1st–8th place). Anthropometric characteristics, maturity status, and body composition variables significantly influenced the final ranking of the tournament. Further studies are needed to better evaluate this relationship in volleyball.

## Figures and Tables

**Figure 1 jfmk-08-00162-f001:**
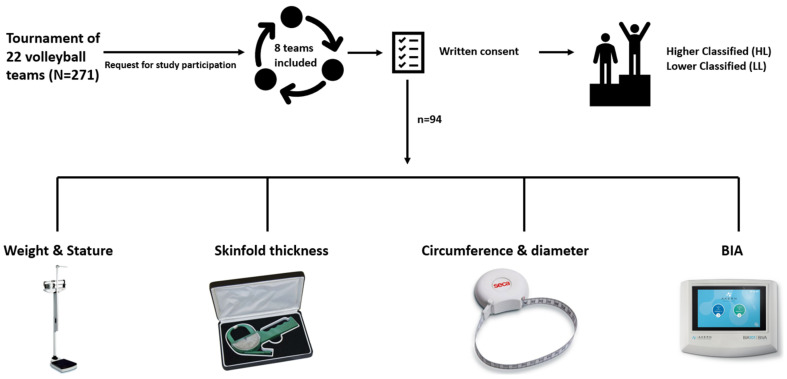
Study design.

**Figure 2 jfmk-08-00162-f002:**
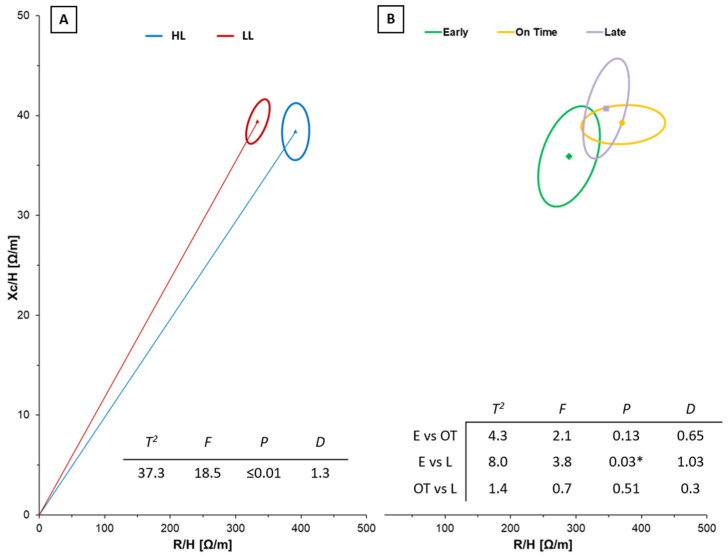
Paired graphs for the multivariate changes in classic resistance and reactance are shown depending on the ranking (**A**) and the maturity status (**B**). The mean vector displacements with 95%, confidence ellipses, and results of Hotelling’s T^2^ test are shown. E = early, OT = on time, L = late, HL= higher level, LL= lower level, * = statistical significant.

**Figure 3 jfmk-08-00162-f003:**
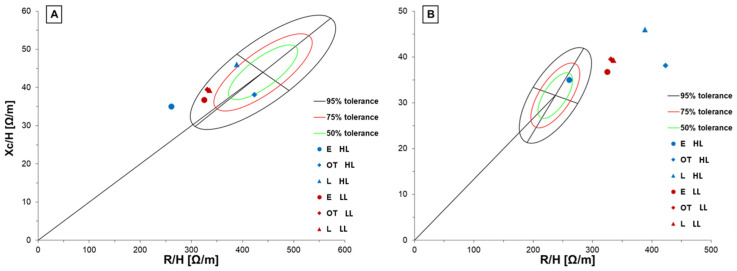
BIVA graphs for the multivariate changes in classical resistance and reactance are shown. The bioimpedance data are plotted on the tolerance ellipses of the general adolescent reference population (**A**) and of the elite volleyball players population (**B**). E = early, OT = on time, L = late, HL= higher level, LL= lower level.

**Table 1 jfmk-08-00162-t001:** Prevalence of maturity status among teams classified as better and worse.

MS (Z ± 1)	Ranking Frequency		∆ Ranks		
	HL	LL	Z or χ^2^	*p*	RR
E	7	5	1.529	0.126	2.256
OT	25	37	0.562	0.574	1.089
L	4	16	−1.901	0.05 *	0.403
Total	36	58	4.98	0.083	

Note. MS = maturity status, E = early, OT = on time, L = late, Z = the test of proportion Z, χ^2^ = Pearson chi-squared test; *p* = *p*-value; RR = risk ratio; *, statistically significant; ∆ difference.

**Table 2 jfmk-08-00162-t002:** General variable statistics according to MS ± 1 year and the final ranking of the tournament.

	HL			LL								
	E (*n* = 7)	OT (*n* = 25)	L (*n* = 4)	E (*n* = 5)	OT (*n* = 37)	L (*n* = 16)	Ranking		MS		Ranking * MS	
Variable	Mean (±SD)	Mean (±SD)	Mean (±SD)	Mean (±SD)	Mean (±SD)	Mean (±SD)	F _(1, 88)_	P	F _(2, 88)_	P	F _(2, 88)_	P
Age (year) #	12.49 (0.81)	12.01 (0.37)	12.75 (0.51)	11.68 (1.82)	13.04 (0.26)	12.40 (0.85)	0.354	0.552	2.865	0.239	3.680	<0.001 *
Weight (Kg)	64.07 (7.97)	52.32 (9.29)	38.00 (4.00)	59.40 (13.99)	45.78 (7.99)	49.88 (9.19)	0.010	0.931	14.540	<0.001 *	1.910	0.154
Stature (cm)	175.89 (7.29)	161.79 (5.30)	148.98 (3.48)	162.36 (16.84)	155.83 (6.71)	159.91 (8.83)	1.580	0.212	13.330	<0.001 *	4.970	0.001 *
Trunk height (cm)	86.81 (3.21)	79.24 (2.65)	70.80 (1.60)	85.68 (8.30)	74.15 (1.46)	80.35 (4.24)	1.130	0.291	43.020	<0.001 *	1.130	0.327
Leg length (cm)	89.07 (5.30)	82.55 (3.74	78.18 (2.32))	76.68 (8.95)	81.68 (6.83)	79.56 (6.39)	5.720	0.019 *	0.800	0.454	5.710	0.005 *
BMI (kg/m^2^)	20.64 (1.43)	19.99 (3.49)	17.13 (1.74)	22.24 (1.69)	19.18 (3.26)	19.37 (2.42)	1.730	0.192	4.860	0.010 *	1.560	0.216

Note: E = early, OT = on time, L = late, MS = maturity status, SD = standard deviation, F = Snedecor–Fischer statistic test, BMI = body mass index, %F = fat percentage, FM = fat mass, FFM = fat-free mass, R = resistance, Xc = reactance, PA = phase angle, * = statistical significant, # = Mann–Whitney rank-sum test and Kruskal–Wallis rank test.

**Table 3 jfmk-08-00162-t003:** Anthropometric statistics according to MS ± 1 year and the final ranking of the tournament.

	HL			LL								
	E (*n* = 7)	OT (*n* = 25)	L (*n* = 4)	E (*n* = 5)	OT (*n* = 37)	L (*n* = 16)	Ranking		MS		Ranking * MS	
Variable	Mean (±SD)	Mean (±SD)	Mean (±SD)	Mean (±SD)	Mean (±SD)	Mean (±SD)	F _(1, 88)_	P	F _(2, 88)_	P	F _(2, 88)_	P
str. Arm circ. (cm)	24.97 (2.68)	20.98 (2.12)	22.48 (2.18)	26.90 (2.72)	22.93 (2.96)	23.60 (2.54)	5.420	0.022 *	8.460	<0.001 *	0.230	0.796
con. Arm circ. (cm)	26.66 (2.34)	22.00 (1.92)	23.85 (2.34)	27.76 (2.40)	23.71 (3.01)	24.61 (2.68)	2.610	0.110	9.230	<0.001 *	0.180	0.832
Calf circ. (cm)	35.69 (1.99)	29.88 (0.38)	32.61 (2.44)	35.06 (3.77)	31.33 (3.65)	32.32 (2.40)	0.060	0.813	9.890	<0.001 *	0.630	0.533
Waist circ. (cm)	70.24 (6.69)	60.55 (1.97)	65.90 (5.87)	73.82 (5.82)	64.74 (6.42)	66.19 (5.22)	2.800	0.098	8.740	<0.001 *	0.860	0.428
Hip circ. (cm)	89.89 (6.60)	71.95 (7.48)	82.62 (6.51)	91.16 (8.20)	80.32 (7.21)	82.72 (6.27)	2.950	0.089	14.240	<0.001 *	2.030	0.137
Humeral diamet. (mm) #	6.79 (0.59)	6.15 (0.44)	6.34 (0.38)	6.50 (0.44)	6.62 (1.47)	6.43 (0.39)	0.001	0.991	6.015	0.050 *	0.740	0.595
Femoral diamet. (mm) #	9.50 (0.58)	8.10 (0.48)	9.40 (1.28)	8.56 (0.85)	8.35 (0.60)	8.59 (0.55)	17.366	<0.001 *	12.647	0.001 *	5.610	<0.001 *
Triceps SK (mm)	10.14 (3.53)	9.25 (2.06)	12.44 (4.3)	16.40 (1.52)	12.31 (4.88)	11.68 (4.12)	5.680	0.019 *	1.160	0.318	3.950	0.023 *
Subscapular SK (mm)	9.29 (2.69)	6.00 (2.16)	10.40 (4.61)	11.20 (1.79)	9.94 (3.02)	8.82 (2.89)	4.400	0.039 *	2.870	0.062	5.600	0.005 *
Supraspinal SK (mm)	11.71 (5.68)	6.50 (2.65)	12.80 (7.05)	15.00 (3.32)	11.06 (4.37)	10.22 (5.10)	2.850	0.095	3.010	0.054	3.970	0.022 *
Suprailiac SK (mm)	13.00 (6.32)	9.25 (3.69)	14.40 (6.84)	16.60 (2.30)	14.31 (5.87)	11.95 (5.34)	2.510	0.117	1.150	0.320	3.350	0.039 *
Medial calf SK (mm)	10.86 (4.02)	11.25 (1.71)	12.68 (4.43)	15.60 (1.52)	13.44 (3.61)	12.65 (3.81)	4.440	0.038 *	0.170	0.846	2.060	0.133
Lateral calf SK (mm)	10.57 (2.15)	12.25 (1.26)	12.80 (3.54)	15.60 (1.34)	12.88 (3.40)	12.97 (2.87)	5.100	0.026 *	0.100	0.909	3.140	0.048 *
TUA (cm^2^)	50.11 (10.41)	35.28 (7.22)	40.56 (7.90)	58.05 (11.83)	42.50 (10.82)	44.81 (9.81)	5.680	0.019 *	8.790	<0.001 *	0.260	0.768
UMA (cm^2^)	38.11 (7.77)	26.13 (4.67)	27.66 (5.07)	38.11 (9.56)	29.25 (6.34)	31.88 (6.06)	1.950	0.166	9.140	<0.001 *	0.680	0.508
UFA (cm^2^)	12.00 (4.77)	9.15 (2.78)	12.91 (5.02)	19.94 (2.90)	13.25 (6.26)	12.93 (5.58)	6.450	0.013 *	2.710	0.072	2.810	0.066
UFI (%)	23.59 (6.36)	25.59 (3.49)	31.20 (8.32)	34.82 (4.14)	30.10 (8.55)	28.15 (7.12)	3.980	0.049 *	0.320	0.729	5.100	<0.001 *
TCA (cm^2^)	101.61 (11.08)	71.03 (1.80)	85.09 (12.66)	98.72 (20.35)	79.11 (17.11)	83.58 (12.16)	0.100	0.749	10.940	<0.001 *	0.730	0.483
CMA (cm^2^)	67.23 (12.97)	40.31 (3.46)	48.74 (10.7)	51.62 (14.34)	43.09 (10.46)	47.33 (9.14)	2.670	0.106	9.480	<0.001 *	2.880	0.061
CFA (cm^2^)	34.37 (8.26)	30.72 (1.80)	36.34 (10.83)	47.10 (7.34)	36.02 (10.22)	36.25 (8.98)	4.340	0.040 *	1.700	0.189	1.910	0.155
CFI (%)	34.06 (8.24)	43.31 (3.46)	42.57 (9.88)	48.32 (4.88)	45.64 (7.99)	43.27 (7.93)	5.980	0.016 *	0.470	0.624	3.240	0.044 *

Note: E = early, OT = on time, L = late, MS = maturity status, SD = standard deviation, F = Snedecor–Fischer statistic test, BMI = body mass index, circ = circumferences, str= stretched, con = contracted, SK = skinfold thickness, TUA = total upper area, UMA = upper muscle area, UFA = upper fat area, UFI = upper fat index, TCA = total calf area, CMA = calf mass area; CFA = calf fat area, CFI = calf fat index, * = statistical significant, # = Mann–Whitney rank-sum test and Kruskal–Wallis rank test

**Table 4 jfmk-08-00162-t004:** Body composition statistics according to MS ± 1 year and the final ranking of the tournament.

	HL			LL								
	E (*n* = 7)	OT (*n* = 25)	L (*n* = 4)	E (*n* = 5)	OT (*n* = 37)	L (*n* = 16)	Ranking		MS		Ranking * MS	
Variable	Mean (±SD)	Mean (±SD)	Mean (±SD)	Mean (±SD)	Mean (±SD)	Mean (±SD)	F _(1, 88)_	P	F _(2, 88)_	P	F _(2, 88)_	P
%F	18.54 (5.21)	14.79 (3.88)	21.23 (6.65)	25.56 (1.95)	20.87 (5.97)	19.40 (5.35)	6.510	0.012 *	2.160	0.121	4.900	0.010 *
FM (kg)	11.94 (3.79)	5.71 (2.05)	11.54 (5.49)	15.29 (4.09)	9.85 (4.06)	9.90 (3.99)	3.350	0.071	6.980	0.001 *	3.780	0.027 *
FFM (kg)	52.13 (6.88)	32.29 (2.32)	40.78 (5.13)	44.11 (10.05)	35.93 (5.05)	39.98 (6.70)	0.690	0.410	14.450	<0.001 *	2.590	0.081
R (Ω)	458.14 (50.11)	578.60 (74.45)	505.38 (54.37)	526.92 (43.87)	520.34 (71.54)	529.48 (63.68)	0.440	0.510	2.670	0.075	3.520	0.034 *
Xc (Ω)	61.66 (12.79)	68.63 (18.91)	61.72 (7.59)	59.70 (5.15)	61.21 (8.97)	63.21 (9.55)	0.570	0.453	0.460	0.631	0.840	0.437
PA #	7.73 (1.93)	6.50 (0.97)	6.74 (0.83)	6.48 (0.39)	6.61 (0.85)	6.78 (0.70)	0.318	0.572	3.022	0.221	1.880	0.106
R/H (Ω/cm)	260.88 (30.86)	388.98 (55.30)	428.79 (582.32)	329.06 (57.49)	335.26 (53.83)	333.65 (55.06)	0.100	0.758	0.390	0.681	0.350	0.708
Xc/H (Ω/cm)	34.89 (5.83)	46.03 (12.45)	38.18 (4.77)	37.20 (6.07)	39.45 (6.77)	39.74 (6.90)	0.240	0.627	3.270	0.043 *	2.180	0.119

Note: E = early, OT = on time, L = late, MS = maturity status, SD = standard deviation, F = Snedecor–Fischer statistic test, %F = fat percentage, FM = fat mass, FFM = fat free mass, R = resistance, Xc = reactance, PA = phase angle, * = statistical significant, # = Mann–Whitney rank-sum test and Kruskal–Wallis rank test.

## Data Availability

Data available on request due to restrictions eg privacy or ethical.

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
