# Peer review of "Differences in Body Composition and Maturity Status in Young Male Volleyball Players of Different Levels"

_jfmk, 2023, doi:10.3390/jfmk8040162_

Round 1

Reviewer 1 Report

Comments and Suggestions for Authors

In this manuscript, the authors investigated the differences in body composition and maturity status in young volleyball male players.

In order for the manuscript to be considered for publication, a major revision as follows is required.

1.     Title: Please clearly state that only male players were investigated in this study.

2.     Abstract: Please revise the statements “The present study aimed to observe differences in anthropometric characteristics and body composition according to the maturity status of the young players of eight volleyball teams.”

Please clearly state in the study aim section of the abstract that only male players were investigated, not just state in the results section of the abstract.

Additionally, please explain why only male players were included in this study. Does gender affect the results?

3.     Line 49: “This is a cross-sectional study design assessed between the 17th and 18th of June 2022, during the National Tournament”.

1)    Is a cross-sectional study design appropriate for this study? The two-day volleyball tournament is subject to serendipity, and it may not represent a long-term youth volleyball tournament. Factors such as the physical and mental state of the players and the technical-tactical level of the coach on that specific day can directly impact the outcome of the game. It is recommended to conduct a long-term study.

2)    Only 6 players on the court in each team. Does each player get the same amount of time on the court? A different amount of time on the court for each player will also affect the outcome of the game. The amount of time on the court for each player should be included in this study. 

4.     Line 54: “A total of 94 young volleyball players were evaluated”. Please clearly state in the “2.1 Participants and Study Design” section that only male players were investigated in this study.

5.     Line 54: “A total of 94 young volleyball players were evaluated”. The sample size is insufficient, making the results less reliable when extrapolated to a larger population. Please explain. Larger sample size is recommended.

6.     Only thirteen-year-old players were included in this study. What about young players of different ages? 11 years old, 12 years old or 14 years old, 15 years old? Can we expect the same results? It is important to consider including young participants of different ages to determine if similar results can be expected across different age groups.

7.     Line 60-64: “each player trained for about 6 hours per week.” But you can not ensure that training content is the same for each team. Please include information about the number of years of training for each player. The same for each player? Long-term study is recommended.

8.     Line 63: “No diet information 63 was collected.” But diet information has a direct impact on the body composition of young players.

9.     Line 63: “whereas 45 minutes looked for technical-tactical skills.” For each training unit, half amount of time was used for technical-tactical skills, which means the technical-tactical level of the coach will also have a direct impact on the outcome of the game. Different teams may have different technical-tactical skills. The relationship between maturity status and the outcome of the game is recommended to be reconsidered.

10.  For Table 1 and Lines 232-234: “In the teams that achieved the higher position of ranking, only four boys were classified as late, while in the teams ranked between the lower level, there were 16 of them.”

1)    There were 16 boys who were classified as late in the teams ranked lower level, but if these players played a little time or even did not play, we cannot conclude that the relationship between maturity status and the outcome of the game, considering only 6 players on the court in each team for the game and only a two-day volleyball tournament. Different players may have different amounts of time on the court. Please consider the effect of the playing time.

2)    Also, the lower ranked teams and the higher ranked teams had similar numbers of players who matured early. Does this mean that maturity status may not be the major factor in determining the outcome of the game? On the contrary, does the technical-tactical level of the coach on that day or other factors may have a significant impact on the outcome of the game?

Author Response

Reviewer 1

In this manuscript, the authors investigated the differences in body composition and maturity status in young volleyball male players.

In order for the manuscript to be considered for publication, a major revision as follows is required.

  1. Title: Please clearly state that only male players were investigated in this study.

Answer (A): The indication was added.

  1. Abstract: Please revise the statements “The present study aimed to observe differences in anthropometric characteristics and body composition according to the maturity status of the young players of eight volleyball teams.”

A: The statement was fixed.

Please clearly state in the study aim section of the abstract that only male players were investigated, not just state in the results section of the abstract.

A: The information was added.

Additionally, please explain why only male players were included in this study. Does gender affect the results?

A: Gender has an important impact on body composition, in fact, different equations are used to evaluate this important aspect, according to gender and age. Females and males have different maturity status according to age. Usually, girls have an earlier maturation than boys. In addition, different methods are usually used to observe the maturity status of the two genders. It could be an interesting study to compare young females and males volleyball players of the same age, but it was not the goal of the present study.

  1. Line 49: “This is a cross-sectional study design assessed between the 17th and 18th of June 2022, during the National Tournament”.

1)    Is a cross-sectional study design appropriate for this study? The two-day volleyball tournament is subject to serendipity, and it may not represent a long-term youth volleyball tournament. Factors such as the physical and mental state of the players and the technical-tactical level of the coach on that specific day can directly impact the outcome of the game. It is recommended to conduct a long-term study.

A: Thank you for the suggestions. The idea of doing a long-term study is an interesting solution to consider. Unfortunately, it was not possible for the present tournament, but for future study, we are thinking about this possibility. In addition, the term was fixed.

2)    Only 6 players on the court in each team. Does each player get the same amount of time on the court? A different amount of time on the court for each player will also affect the outcome of the game. The amount of time on the court for each player should be included in this study. 

A: Unfortunately, we did not have the possibility to collect the amount of time on the court for each player. This information was added in the limitations part (lines 333-347). However, all the teams settled the line up to win the competition, so the amount of time spent on the court for each team player was similar. We think this parameter could not affect the body composition means differences between the teams.

  1. Line 54: “A total of 94 young volleyball players were evaluated”. Please clearly state in the “2.1 Participants and Study Design” section that only male players were investigated in this study.

A: The clarification was added.

  1. Line 54: “A total of 94 young volleyball players were evaluated”. The sample size is insufficient, making the results less reliable when extrapolated to a larger population. Please explain. Larger sample size is recommended.

A: Thank you for the suggestion. We know that a larger sample is recommended to have more reliable results, but unfortunately, not all the team decided to participate in the present study.

  1. Only thirteen-year-old players were included in this study. What about young players of different ages? 11 years old, 12 years old or 14 years old, 15 years old? Can we expect the same results? It is important to consider including young participants of different ages to determine if similar results can be expected across different age groups.

A: This information is in the limitation of the study. There is no doubt that a larger study including different ages and different genders could be more interesting and could have a higher influence on literature, but the national tournament was only for thirteen-year-old male players. In Italy there are no national tournaments for 11 or 12-year-old male players, there are only provincial tournaments, for example. Maturity status is an important discriminating variable until the achievement of complete physical development, so future study could analyze different ages, such as 14-year-old or 15-year-old volleyball players.

  1. Line 60-64: “each player trained for about 6 hours per week.” But you can not ensure that training content is the same for each team. Please include information about the number of years of training for each player. The same for each player? Long-term study is recommended.

A: We did not ask information about the years of training for each player. We add this information in the limitation of the study (lines 333-347)

  1. Line 63: “No diet information 63 was collected.” But diet information has a direct impact on the body composition of young players.

A: This is one of the limitations of the present study that we add in the limitation part (lines 333-347).

  1. Line 63: “whereas 45 minutes looked for technical-tactical skills.” For each training unit, half amount of time was used for technical-tactical skills, which means the technical-tactical level of the coach will also have a direct impact on the outcome of the game. Different teams may have different technical-tactical skills. The relationship between maturity status and the outcome of the game is recommended to be reconsidered.

A:  Thank you for the suggestion. It is true that different coaches could teach in different ways and with different methods, but it is also true that volleyball is a team sport in which the anthropometric characteristics have a fundamental role, due to the presence of the net, which in the category under 13 is 2.15m high. It could be very interesting investigating how maturity status, volleyball skills and body growth correlate. This study is just a preliminary study, and we hope we could investigate on further aspects.

  1. For Table 1 and Lines 232-234: “In the teams that achieved the higher position of ranking, only four boys were classified as late, while in the teams ranked between the lower level, there were 16 of them.”

1)    There were 16 boys who were classified as late in the teams ranked lower level, but if these players played a little time or even did not play, we cannot conclude that the relationship between maturity status and the outcome of the game, considering only 6 players on the court in each team for the game and only a two-day volleyball tournament. Different players may have different amounts of time on the court. Please consider the effect of the playing time.

A: Thank you for the suggestion. Unfortunately, we did not have the possibility of having an amount of the time for each player in the court. We put this aspect in the limitation of the present study.

2)    Also, the lower ranked teams and the higher ranked teams had similar numbers of players who matured early. Does this mean that maturity status may not be the major factor in determining the outcome of the game? On the contrary, does the technical-tactical level of the coach on that day or other factors may have a significant impact on the outcome of the game?

A:  Thank you for the interesting suggestion. It is also true that the lower-ranked teams presented a higher number of players who matured late, and, in our opinion, this aspect influenced the final ranking, more than the technical-tactical level of the coach on that day.

Reviewer 2 Report

Comments and Suggestions for Authors

The authors compare the state of maturity and anthropometric variables in volleyball players, taking into account the ranking of the teams. I think that the information extracted in the results is not sufficiently explained and the discussion is superficial.

A few specific aspects.

1.- In section 2.1 participants, the authors should indicate that all participants are male.

2.- In the hypothesis, the authors do not indicate the meaning of the differences. They say: “It was hypothesized that players who reached a higher position in the ranking would exhibit differences in maturity status and their anthropometric and body composition profile”. The authors should state what differences they think they will find.

3.-  Table 2 is huge. The authors could divide it into several tables, grouping the variables (general, anthropometric, bioimpedance) and/or eliminate variables that are inconsequential or of little interest for the objective of the study.

4.- Why do the authors only use references to football teams? There are works done on basketball players (https://doi.org/10.1590/1980-0037.2019v21e60248; | https://doi.org/10.1038/s41598-021-01401-4o) or handball. ( https://doi.org/10.3390/app13053012; DOI: 10.1055/s-0031-1298000) who might have an interest in the comparison.

5.- The authors do not indicate the years of experience of the players, nor whether there are differences between teams or groups related to experience. The different position in the rankings could be due to the fact that the better teams have more experienced players than the eliminated teams, regardless of maturity. If the authors have not taken this into account, this is a limitation.

6.- The authors should explain why, in the high-level group, the "OT" boys have strikingly lower values than the other two groups in determining variables. This one is especially striking with those in the "L" group. For example, weight, height, trunk length. It is observed that these boys (OT) are smaller than those with delayed maturation.

7.- If the objectives of the study are:  "(a) compare the prevalence of maturity status among volleyball players of the teams that have reached different positions in the ranking of a national tournament, and (b) investigate the relationship between maturity status and anthropometric, performance, body composition parameters and BIVA". The conclusions should be the response to those objectives and not provide other comments or ideas.

I cross out what are not conclusions from your study.

5. Conclusions 311

Evaluating maturity status is an important aspect to consider in talent selection and the research connected to this aspect. Maturation has a great influence on body composition changes and physical performance. Literature about this topic is constantly increasing in many sports such as soccer, but there is a lack of studies about volleyball, especially among adolescent male volleyball players. In the present study, young volleyball players classified as early had higher values of the anthropometric characteristics linked to better performance (represented by the final ranking of the tournament). In fact, among the eight teams, two of them that presented the most early maturing boys were ranked in the top places of the tournament (1st -8th place). The results of the present study could have practical implications for talent selection, but further studies are needed to better evaluate the  relationship between maturity status, body composition and performance in volleyball.

8.- In introduction, discussion and conclusions, there are ideas that are repeated several times:

1.                   No studies or more studies needed: Lines: 12, 36, 223, 250, 266, 288, 295, 297. It is repetitive and boring for the reader.

2.                   Talent detection. Lines 22, 226, 312, 321. It is not part of the objectives and therefore should not be included in the conclusion.

9.- References. There are errors in some references. Others are incomplete, e.g. references: 6, 10, 18. In some citations there is the full name of the journal and in others the abbreviations.

Author Response

Review 2

The authors compare the state of maturity and anthropometric variables in volleyball players, taking into account the ranking of the teams. I think that the information extracted in the results is not sufficiently explained and the discussion is superficial.

A few specific aspects.

1.- In section 2.1 participants, the authors should indicate that all participants are male.

Answer (A): Thank you for the suggestion. The information was added.

2.- In the hypothesis, the authors do not indicate the meaning of the differences. They say: “It was hypothesized that players who reached a higher position in the ranking would exhibit differences in maturity status and their anthropometric and body composition profile”. The authors should state what differences they think they will find.

A: The authors’ hypotheses were added in lines 46-49.

3.-  Table 2 is huge. The authors could divide it into several tables, grouping the variables (general, anthropometric, bioimpedance) and/or eliminate variables that are inconsequential or of little interest for the objective of the study.

A: The table was divided and fixed.

4.- Why do the authors only use references to football teams? There are works done on basketball players (https://doi.org/10.1590/1980-0037.2019v21e60248; | https://doi.org/10.1038/s41598-021-01401-4o) or handball. ( https://doi.org/10.3390/app13053012; DOI: 10.1055/s-0031-1298000) who might have an interest in the comparison.

A: Thank you for the suggestions, we added a comparison also with these two sports.

5.- The authors do not indicate the years of experience of the players, nor whether there are differences between teams or groups related to experience. The different position in the rankings could be due to the fact that the better teams have more experienced players than the eliminated teams, regardless of maturity. If the authors have not taken this into account, this is a limitation.

A: Thank you for the suggestion. Unfortunately, it was not possible to collect this kind of information at the tournament, so we added the lack of information about the years of experience as a limitation of the present study (lines 333-347).

6.- The authors should explain why, in the high-level group, the "OT" boys have strikingly lower values than the other two groups in determining variables. This one is especially striking with those in the "L" group. For example, weight, height, trunk length. It is observed that these boys (OT) are smaller than those with delayed maturation.

A: Thank you so much for the signalling. It was an error in the creation of the table, now it is fixed.

7.- If the objectives of the study are:  "(a) compare the prevalence of maturity status among volleyball players of the teams that have reached different positions in the ranking of a national tournament, and (b) investigate the relationship between maturity status and anthropometric, performance, body composition parameters and BIVA". The conclusions should be the response to those objectives and not provide other comments or ideas.

I cross out what are not conclusions from your study.

  1. Conclusions 311

Evaluating maturity status is an important aspect to consider in talent selection and the research connected to this aspect. Maturation has a great influence on body composition changes and physical performance. Literature about this topic is constantly increasing in many sports such as soccer, but there is a lack of studies about volleyball, especially among adolescent male volleyball players. In the present study, young volleyball players classified as early had higher values of the anthropometric characteristics linked to better performance (represented by the final ranking of the tournament). In fact, among the eight teams, two of them that presented the most early maturing boys were ranked in the top places of the tournament (1st -8th place). The results of the present study could have practical implications for talent selection, but further studies are needed to better evaluate the  relationship between maturity status, body composition and performance in volleyball.

A: Conclusions were fixed according to the suggestions.

8.- In introduction, discussion and conclusions, there are ideas that are repeated several times:

  1. No studies or more studies needed: Lines: 12, 36, 223, 250, 266, 288, 295, 297. It is repetitive and boring for the reader.

A: Thank you for the suggestion, these parts were removed.

  1. Talent detection. Lines 22, 226, 312, 321. It is not part of the objectives and therefore should not be included in the conclusion.

 A: Thank you for the suggestion, in line 43 we added why in our opinion talent selection is part of the present study.

9.- References. There are errors in some references. Others are incomplete, e.g. references: 6, 10, 18. In some citations there is the full name of the journal and in others the abbreviations.

A: Thank you for the suggestion, the references were fixed.

Round 2

Reviewer 1 Report

Comments and Suggestions for Authors

Please submit a revised manuscript without annotations.

Reviewer 2 Report

Comments and Suggestions for Authors

I note that you have taken my comments into account.

Best regards